# Study of New Mini-Channel Trans-Critical CO_2_ Heat Pump Gas Cooler

**DOI:** 10.3390/mi13081206

**Published:** 2022-07-29

**Authors:** Jiawei Jiang, Shiqiang Liang, Can Ji, Longyan Wang, Chaohong Guo

**Affiliations:** 1Research Center of Fluid Machinery Engineering and Technology, Jiangsu University, Zhenjiang 212013, China; 2212011015@stmail.ujs.edu.cn (J.J.); longyan.wang@connect.qut.edu.au (L.W.); 2Institute of Engineering Thermophysics, Chinese Academy of Sciences, Beijing 100190, China; guochaohong@iet.cn; 3School of Engineering Science, University of Chinese Academy of Sciences, Beijing 100049, China; 4Dalian National Laboratory for Clean Energy, Dalian 116000, China; 5Energy Research Institute, Qilu University of Technology (Shandong Academy of Sciences), Jinan 250014, China; jic@sderi.cn

**Keywords:** trans-critical carbon dioxide heat pump gas cooler, spiral heat exchanger, numerical simulation

## Abstract

A gas cooler is one of the important parts of a carbon dioxide (CO_2_) heat pump water heater, and it must meet the needs of not only pressurization but also heat transfer. It is important to study gas coolers. In this paper, a heat exchanger with a spiral channel is studied. ANSYS CFX software was used to analyze the flow and heat transfer characteristics of the heat exchanger (single-plate model). The influences of the cooling pressure of CO_2_, the mass flux of CO_2_, the mass flux of water and the channel radius of CO_2_ are discussed. In this paper, the results show that the cooling pressure of CO_2_, the mass flux of CO_2_ and the channel radius of CO_2_ all have a large influence on the local heat transfer coefficient: with an increase in the cooling pressure of CO_2_, the peak value of the heat transfer coefficient of CO_2_ decreases and the average heat transfer coefficient decreases; with an increase in the mass flux of CO_2_, the peak value of the heat transfer coefficient of CO_2_ increases and the average heat transfer coefficient increases; and with a decrease in the channel radius of CO_2_, the peak value of the heat transfer coefficient of CO_2_ increases. The water mass flux has only a slight effect on heat transfer, and the lower cooling pressure of CO_2_ corresponds to a higher peak heat transfer coefficient, which can reach 27.5 kW∙m^−2^∙K^−1^ at 9 MPa.

## 1. Introduction

CO_2_ is an excellent candidate due to its nontoxicity, incombustibility, safety, low cost and environmentally benign nature (ODP = 0, GWP = 1). It is widely implemented in refrigeration and heat pump systems, air conditioning and various industrial uses. Lorentzen first proposed the application of trans-critical carbon dioxide cycles to automotive heat pumps in 1993, and this was the beginning of the widespread application of carbon dioxide heat pumps in various fields [1]. The gas cooler is one of the important parts of a CO_2_ heat pump. According to the different cooling media used, gas coolers are mainly divided into air-cooled and water-cooled. The air-cooled type is mainly used for automotive heat pumps, divided into tube-fin and microchannel coolers; the water-cooled type is mainly used for trans-critical CO_2_ heat pump water heaters, mostly using double pipes. Chang analyzed tube-fin gas coolers with different circuits through an experimental study, as well as the effect of geometric factors on heat transfer, and found that three circuits were optimal [2]. Chai simulated a tube-fin gas cooler with distributed modeling and used the ε-NTU method to research the effect of design factors on the cooling performance of the whole gas cooler [3]. Wang studied a herringbone wavy-fin gas cooler and proposed a new heat transfer correlation with an average error of 8.82% [4]. Ge first constructed a mathematical model by dividing the tube-fin gas cooler into four regions and verified it with experimental results, which matched well; then, a finite difference model was developed to simulate the performance of the gas cooler; finally, the model was used to simulate and analyze the gas cooler with different circuits [5,6,7]. Zhang analyzed the heat transfer performance of an existing tube-fin gas cooler model with the CFD method [8,9]. Zilio and Simone designed three different shell and tube gas coolers and performed experimental and simulation analyses; they found that the influence of different tube types of inlet water flow rates on the heat production rate was reduced at 10 MPa, and the heat production rate was relatively high for smooth and internally grooved tubes [10]. Fronk designed a cross-flow plate type microchannel gas cooler and researched the heat transfer and pressure drop characteristics of the gas cooler with both five water flow channels and with seven water flow channels, at different inlet flow rates and inlet temperatures, and he summarized the heat transfer correlations based on the experimental results and also simulated the performance of a microchannel heat exchanger with 12 water flow channels. The study showed that the microchannel gas cooler was superior to the double pipe gas cooler [11,12]. Starace proposed a method to investigate and find the overall performance of a heat exchanger starting from CFD simulations at the micro-scale; the study showed that the multi-scale approach leads to a better accuracy level than the full-scale one [13]. Fiorentino carried out some numerical simulations of the evaporative condenser heat and mass transfer processes at the tube scale, then studied two different types of flow by varying the water-to-air mass flow ratio. He found that a decrease of 25% in the water-to-air mass flow ratio led to the film separating into droplets. A test rig to investigate the evaporative condenser at small scale had been designed and built. An increase of 37.5% in the air flow rate (at a constant rate of sprayed water) led to a maximum reduction in the heat transfer rate of 50% [14,15]. Bilal used the fouling model to investigate the risk-based thermal performance of these evaporative heat exchangers and carried out sensitivity analysis, finding that the condensing temperature is the most sensitive parameter; for any increase in the inlet’s relative humidity, the normalized sensitivity of the surface area increased [16,17]. Scattina developed a new methodology that allows scaling-up from the CFD analysis of a small element to the prediction of the performance of an entire compact heat exchanger [18].

Current research on trans-critical CO_2_ heat pump gas coolers is focused on the double pipe and tube-fin types, with no structural innovation. Gas coolers must be able to withstand high pressure, with high reliability, low costs, compactness, efficiency and other requirements. A double pipe gas cooler is not compact enough, not easy to scale and is difficult to clean, so it is not good enough to meet the above needs. The concept of a spiral heat exchanger was first proposed in the late nineteenth century and was reinvented in Sweden during the 1930s. Up to now, much research relating to the characteristics of spiral heat exchangers has been presented by researchers. According to Bes, spiral heat exchangers have high thermal efficiency and compactness, and they have only one cross-sectional channel for each fluid, in which the flow direction is constantly changing, thus eliminating the flow stagnation zone channel and making it difficult for fouling to remain, and it rarely scales [19]. If the heat exchanger is fouled, this can be soaked in white vinegar, dissolved with anhydrous ethanol, and finally blown with dry compressed air since the CO_2_ channel must be kept dry. In this paper, a new spiral plate mini-channel gas cooler with asymmetric channels on both sides is proposed for CO_2_ heat pump gas coolers with relatively large water and CO_2_ flow rates. 

ANSYS CFX software (ANSYS, Canonsburg, PA, USA) was used to simulate the flow and heat transfer characteristics of supercritical carbon dioxide in the new mini-channel gas cooler, and to analyze the influence of different structural parameters and working conditions on heat transfer, so as to provide more directions and new ideas for research on trans-critical carbon dioxide heat pump gas coolers.

## 2. Physical Model

The geometric details of the new mini-channel heat exchanger are shown in Figure 1. The mini-channel heat exchanger consists of two different plates alternating in sequence. Figure 1a shows one of them with a semi-circular channel of CO_2_ above and a semi-elliptical channel of CO_2_ below; the other plate has a semi-elliptical channel above and a semi-circular channel below, and the two plates are alternately stacked in sequence to form the whole heat exchanger. The overall schematic is shown in Figure 1b, and the important dimensions of the gas cooler are listed in Table 1. The hydraulic diameters of the CO_2_ channels and the water channels were selected according to plate thicknesses available on the market, pressure-bearing capacity and mass flow rate. CO_2_ and water are counter-flowing, with CO_2_ flowing from inside to outside and water flowing from outside to inside, as seen in Figure 1a. Two different channels are distributed on both sides of the plate, which can prevent the mixing of fluids. Each plate is stacked and combined into a whole, connected together by diffusion welding, and can withstand high pressure. Diffusion welding is a kind of pressure welding and refers to the process of surfaces being brought into mutual contact under the action of high temperature pressure. With the connected surfaces close to each other, and local plastic deformation, after a certain period of time the atoms of the bonding layer diffuse and intersperse with each other, forming a reliable connection of the whole.

## 3. Structural Strength Analysis

As the CO_2_ side of the gas cooler must be able to withstand high pressures of at least 13 MPa, the strength analysis of the designed heat exchanger was first performed to determine whether its pressure bearing capacity meets the requirements. The ANSYS Static Structural test was used for the analysis, and the model was simplified to a straight section of 1 cm in length, as shown in Figure 2. The bottom and right side of the models were set as symmetric boundary conditions, and the front face was set as frictionless support. After setting up the conditions, the stress and strain solutions were carried out. The stress and strain are displayed in Figure 3. Figure 3 shows the results with the boundary conditions of 20 MPa applied on the CO_2_ side and 0.5 MPa applied on the water side. It was found that the elliptical channel can bear 20 MPa pressure, which was far beyond the pressurization.

## 4. Numerical Model

### 4.1. Data Reduction

The relevant governing equations for the turbulence model are shown in Appendix A. The local bulk temperature of the supercritical CO_2_ and the cooling water, the inner wall temperature of the channel and the heat flux on the wall along the channel were obtained from the numerical results. The heat transfer coefficient of the supercritical CO_2_ flowing in the channel was calculated as:
(1)
$$h=\frac{q_{x}}{T_{b,CO_{2}}-T_{W,x}}$$

where *q_x_* and *T_W,x_* are the circumferentially averaged heat flux and temperature on the inner wall along the channel, respectively, which were calculated as:
(2)
$$q_{x}=\frac{\int_{L}^{}qdL}{L}$$


(3)
$$T_{W,x}=\frac{\int_{L}^{}TdL}{L}$$


*L* is the length of the semicircle arc on the inner wall where the axial position is equal to *x*. *x* represents the axial location along the channel. Unlike the traditional definition of the bulk temperature of fluids with constant properties, the bulk temperature of the supercritical CO_2_ was determined by the average enthalpy, which was calculated as:
(4)
$$H_{b}=\frac{\int \rho uHdA}{\int \rho udA}$$


The bulk temperature of the supercritical CO_2_ was defined as:
(5)
$$T_{{b,\mathrm{CO}}_{2}}=f(H_{b},\;P)$$


The mass flux was defined as:
(6)
$$G=\frac{m}{A}$$


### 4.2. Reliability Verification

Because of the novel structure proposed in this paper, there were no relevant experiments for CO_2_ heat exchangers that could be verified. In Paisarn’s study, numerical and experimental results of the heat transfer and flow characteristics of a horizontal spiral-coil tube (Figure 4) were investigated. The RNG 
k−ε
 two-equation turbulence model was applied to simulate turbulent flow and heat transfer characteristics. Reasonable agreement was obtained by comparison between the results from the experiment and those obtained from the model [20]. The channel of CO_2_ in this paper was a circular cross-section spiral channel, and a similar spiral tube was selected for turbulence model verification. Experimental results for the conjugate heat transfer between supercritical CO_2_ and water by Zhang were used to validate the numerical method [21]. The geometric model of the spiral tube is shown in Figure 5. The specific dimensions of the spiral tube used in the simulation were: inner diameter 9 mm, outer diameter 12 mm, coil diameter 283 mm, pitch 32 mm. Supercritical CO_2_ flowed from the bottom to the top inside the spiral tube and was heated by the wall. The pressure was 8.02 MPa, the inlet mass flux was 97.92 kg/(m^2^·s), the inlet temperature was 288.15 K and the heat flux was 10.4 kW/m^2^. Figure 5 shows the variation in the bulk temperature with *x*/*d* at different cross sections simulated by the two turbulence models compared with the experimental data of Zhang. The horizontal coordinate *x*/*d* in the figure indicates the ratio of the distance from this cross-section to the inlet cross-section to the pipe diameter. It can be seen from Figure 6 that the results obtained from the RNG 
k−ε
 model simulations are in better agreement with the experimental data, indicating that the RNG 
k−ε
 model has higher accuracy in calculating the rotational flow. The large deviation in the two turbulence models in the high temperature prediction may be attributed to the constant heat flow conditions that cause heat to accumulate at the outlet. The RNG 
k−ε
 model was used in the subsequent simulations.

### 4.3. Grid Independence

Due to the large computational resources consumed by the overall analysis, a single plate of the heat exchanger was used for the analysis in this paper, as shown in Figure 1a. The spiral channel of CO_2_ on one side of the plate had a 1 mm radius semicircle cross-section, a spiral start radius of 10 mm, an end radius of 90 mm and a pitch of 8 mm. On the other side, for the spiral channel of water, the long semi-axis of the channel cross-section was 2.5 mm, the short semi-axis was 1.3 mm, the spiral start radius was 14 mm, the end radius was 86 mm and the pitch was 8 mm. The veneer was a 200 mm × 200 mm square.

Workbench meshing was used to divide the fluid domain and the solid domain into meshes, and the cross-sections of two adjacent channels of heat transfer units in a single plate are displayed in Figure 7. To eliminate the impact of the entrance and exit regions, some extensions were made to the channel. In order to obtain more accurate calculation results, and at the same time avoid excessive calculation, it was necessary to compare the calculation results under different grid numbers, and that grid division is shown in Figure 8. The first layer thickness in the CO_2_ region was set to 0.0000003 mm to make sure that the dimensionless wall distance y+ < 1. The mass flow rates of CO_2_ and water were 0.0033 kg·s^−1^ and 0.01 kg·s^−1^, respectively. The inlet temperatures of CO_2_ and water were 373.15 K and 293.15 K, respectively. The pressure of CO_2_ and water were 9 MPa and 0.5 MPa, respectively. Setting symmetric boundary conditions on the surface of the two fluid domains, the residuals reached 10^−6^ and the calculation converged. In this paper, five sets of different grid dimensions with total grid numbers of 3.0 million, 3.7 million, 4.3 million, 5.2 million and 6.0 million were selected to carry out subsequent numerical simulations by calculating the pressure drop of CO_2_, as displayed in Figure 9, and finally, a 5.2 million grid was selected.

## 5. Results and Discussion

### 5.1. Effect of Cooling Pressure of CO_2_ on Heat Transfer

In this section, the main goal is to research the effects of different cooling pressures on heat transfer. The mass flow rates of CO_2_ and water were 0.0033 kg·s^−1^ and 0.01 kg·s^−1^, respectively. The inlet temperatures of CO_2_ and water were 373.15 K and 293.15 K, respectively. The pressure of CO_2_ and water were 10 MPa and 0.5 MPa, respectively. The CO_2_ heat transfer coefficient under cooling pressure is shown in Figure 10. The specific calculation conditions are listed in Table 2.

It can be seen from Figure 10 that the peak value of the local heat transfer coefficient of CO_2_ decreases as the cooling pressure increases, and the critical temperature corresponding to the peak value is different. The fluid temperatures corresponding to the peak heat transfer coefficients at cooling pressures of 9, 10, 11 and 12 MPa were 313.15 K, 320.15 K, 324.15 K and 334.15 K, respectively. The closer to the pseudocritical point, the higher the constant pressure specific heat capacity and thermal conductivity of CO_2_, hence the higher the corresponding local heat transfer coefficient. The reason the heat transfer coefficients are relatively close at different pressures away from the pseudocritical point is that the specific heat capacity of carbon dioxide changes slightly at different pressures and the larger thermodynamic changes are mainly concentrated near the pseudocritical point. The peak heat transfer coefficient of CO_2_ reaches 27.5 kW∙m^−2^∙K^−1^ at a cooling pressure of 9 MPa. Figure 11 shows that the average CO_2_ heat transfer coefficient decreases gradually as the cooling pressure increases. Figure 12 illustrates the pressure distribution of CO_2_.

### 5.2. Effect of Inlet Mass Flux of CO_2_ on Heat Transfer

In this section, the effects of different mass fluxes of CO_2_ on the heat transfer characteristics are discussed. The conditions were the same as in the previous section except for the mass flux of CO_2_. The mass flow rate of water was 0.01 kg·s^−1^. The inlet temperatures of CO_2_ and water were 373.15 K and 293.15 K, respectively. The pressure of CO_2_ and water were 10 MPa and 0.5 MPa, respectively. The specific calculation conditions are given in Table 3.

The variation in the convective heat transfer coefficient with different mass fluxes in the CO_2_ channel is given in Figure 13. As the mass flux increases, the convective heat transfer coefficient of CO_2_ also increases. The reason for this is that as the mass flux increases, the intensity of the turbulence in the channel increases and the fluid continuously scours the wall, affecting the formation of the boundary layer. The changing pattern of the average heat transfer coefficient at different mass fluxes is described in Figure 14. Figure 15 shows the variation in temperature of CO_2_ at different mass fluxes. It can be seen from the figure that the heat transfer in the outer ring is becoming worse and the channel length can be considered to be shortened in the subsequent optimization. Figure 16 and Figure 17 give the variation in the pressure drop in the channel of CO_2_ with the mass flux. With the increase in the mass flux, the pressure drop in the channel increases continuously, and the variation in the pressure drop in the channel at the lower mass flux is slightly smaller than that at the larger mass flux. The difference in the pressure drop tends to increase gradually with the increasing of the mass flux. According to the changing of the pressure drop and the heat transfer coefficient with the mass flux, it is necessary to balance the two conditions to choose the most suitable working condition. In the conventional type of heat exchangers, the average heat transfer coefficient of double pipe and shell and tube heat exchangers is mostly below 5 kW∙m^−2^∙K^−1^, while the average heat transfer coefficient of PCHE and microtube heat exchangers can reach 10 kW∙m^−2^∙K^−1^; the pressure drop is mostly below 100 kpa, so the disadvantage of spiral heat exchangers is the large pressure drop. According to the ngo study, the volumetric thermal capacity of PCHE can reach 25,000 kw/m^3^ with a hydraulic diameter of 1 mm and the volumetric thermal capacity of a spiral heat exchanger is 19,000 kw/m^3^ [22].

Figure 18 shows the effect of different mass fluxes in the water channel on the heat transfer coefficient of CO_2_. From Figure 15, it can be seen that increasing the inlet mass flux on the water side slightly increases the heat transfer coefficient of CO_2_. Due to the heat transfer between the water and the outer surface of the metal wall increasing with the intensity of the water turbulence as the mass flux of the water increases, the heat transfer coefficient between carbon dioxide and the metal wall increases accordingly. The water flow rate affects the outer wall surface temperature of CO_2_, and enhanced heat transfer can be achieved by changing the outer wall surface temperature of the CO_2_. This also corroborates the study of Jiang that proposed that, to enhance heat transfer, the heat transfer of water is an influential factor that needs to be considered comprehensively [23].

### 5.3. Effect of Channel Size of CO_2_ on Heat Transfer

In this section, the effects of different channel radiuses of CO_2_ on the heat transfer characteristics are investigated. The mass flow rates of CO_2_ and water were 0.002466 kg·s^−1^ and 0.01 kg·s^−1^, respectively. The inlet temperatures of CO_2_ and water were 373.15 K and 293.15 K, respectively. The pressure of CO_2_ and water were 10 MPa and 0.5 MPa, respectively. The effects on the heat transfer coefficient of CO_2_ at different channel radiuses are shown in Figure 19.

Figure 19 illustrates the effect of the channel radius on the heat transfer coefficient of CO_2_. It can be seen that the heat transfer coefficient of CO_2_ increases as the channel radius of CO_2_ decreases. The reason is that reducing the channel radius of CO_2_ corresponds to reducing the cross-section, which is equivalent to increasing the mass flux under the same conditions and increasing the turbulence intensity. Furthermore, the increase in the heat transfer coefficient was larger when the channel radius of CO_2_ was 0.8 mm, and this was accompanied by a dramatic increase in the pressure drop. Under this condition, the heat transfer coefficients of CO_2_ with all three different channel radiuses reached the peak at the pseudocritical point. Figure 20 and Figure 21 display the effect of the channel radius on the pressure drop of CO_2_. As the channel radius decreases, the pressure drop increases.

### 5.4. Field Synergy Principle of Heat Exchange

A further analysis of the heat transfer characteristics of CO_2_ used the field synergy principle. This starts from the two-dimensional boundary layer convective heat transfer differential equation. Guo deduced the field synergy principle by considering convection as a special endothermic source [24]. Generally, the most frequently used field synergy principle analysis mainly uses the synergy angle as a primary indicator of synergy degree to analyze the mechanism of convective heat transfer enhancements. The synergy angle can be calculated with the following equation:
(7)
$$\beta =\cos^{-1}(\frac{\left|\vec{U}\cdot \nabla T\right|}{\left|\vec{U}\right|\cdot \left|\nabla T\right|})$$


Figure 22 shows the long-range synergy angle distribution within the channel of CO_2_. The smaller synergy angle is concentrated in the center of the channel, mainly because the temperature gradient at the center of the channel is along the axial direction and the flow direction consistency is higher. Heat transfer from the channel wall is dominated by thermal conduction near the wall boundary layer, so the temperature gradient is mainly along the wall normal direction, with a large angle to the flow direction. The location of the better synergy within the channel of CO_2_ is at the peak heat transfer coefficient (*x*/L = 0.468).

## 6. Conclusions

In this study, the heat transfer between CO_2_ and water in a spiral heat exchanger is numerically analyzed. The main results of this study are as follows:Spiral plate heat exchanger pressurization is good; when the radius of the circular channel is 1 mm, the elliptical channel short axis is 2.3 mm and the long axis is 5 mm, it can withstand the high pressure of 20 MPa.As the cooling pressure decreases, the average heat transfer coefficient of CO_2_ increases, the heat transfer capacity is enhanced and the peak heat transfer increases. Mainly owing to the fact that the lower the pressure, the higher the specific heat will be at constant pressure, the average heat transfer coefficient of CO_2_ is also higher.The higher the mass flux of CO_2_, the higher the corresponding peak local convective heat transfer coefficient, which occurs near the pseudocritical temperature corresponding to the cooling pressure.The effects of changing the mass fluxes of water and CO_2_ on the heat transfer coefficient of CO_2_ are different. The increase in the mass flux of CO_2_ and the decrease in the channel radius have a more obvious effect on the heat transfer coefficient, while the increase in the mass flux of water affects the temperature of the wall surface of CO_2_.The locations with smaller synergy angles within the channel of CO_2_ are distributed in the center of the channel, and the locations with better synergy occur at the location of the peak heat transfer coefficient of CO_2_.

## Figures and Tables

**Figure 1 micromachines-13-01206-f001:**
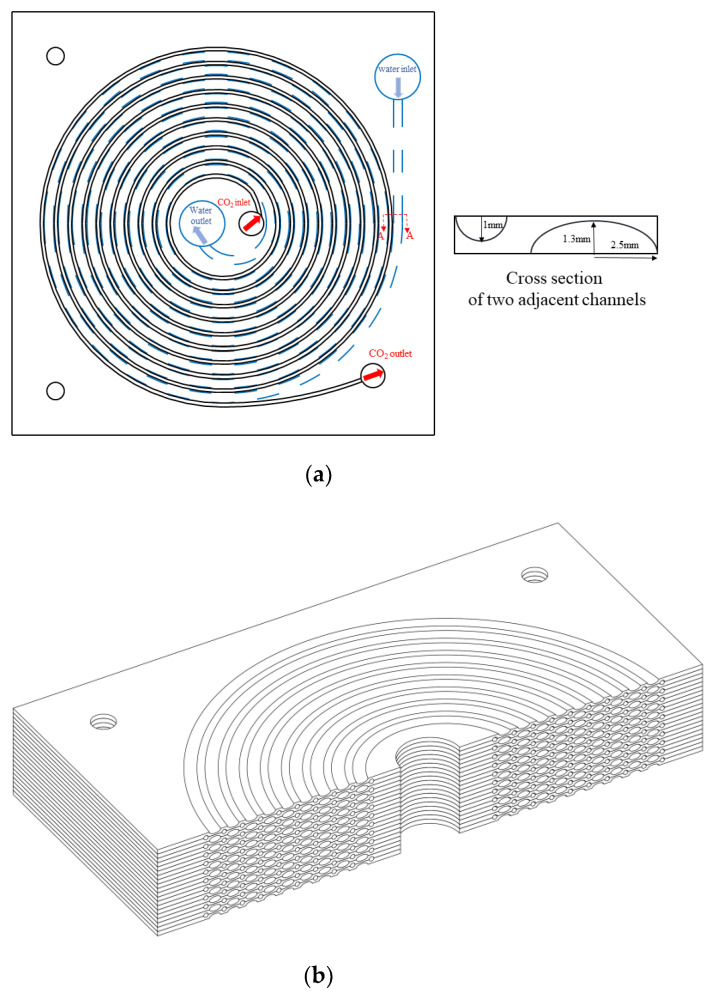
(**a**) A type of structural plate; (**b**) Overall schematic.

**Figure 2 micromachines-13-01206-f002:**
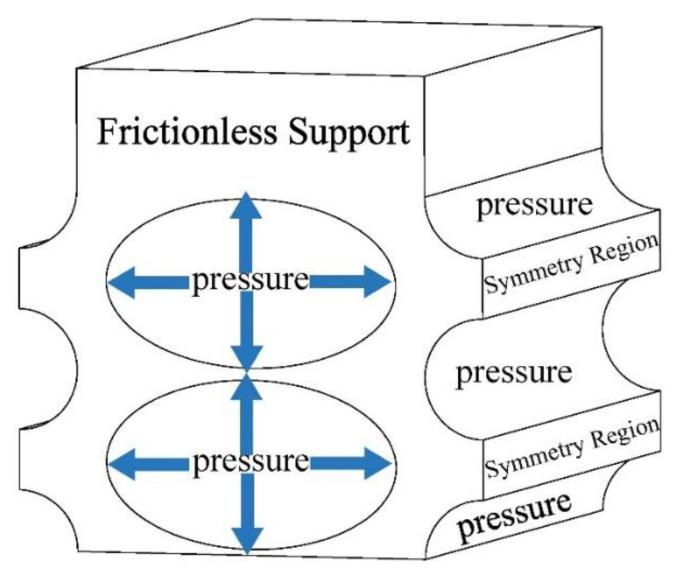
Strength analysis model and boundary condition setting.

**Figure 3 micromachines-13-01206-f003:**
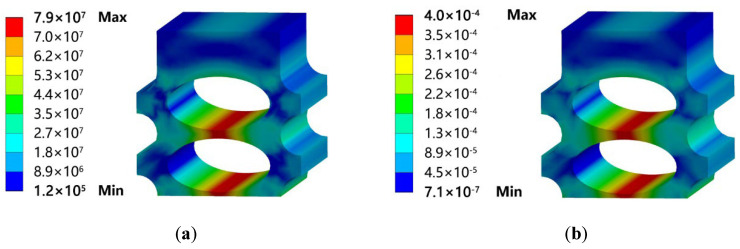
(**a**) Stress; (**b**) Strain.

**Figure 4 micromachines-13-01206-f004:**
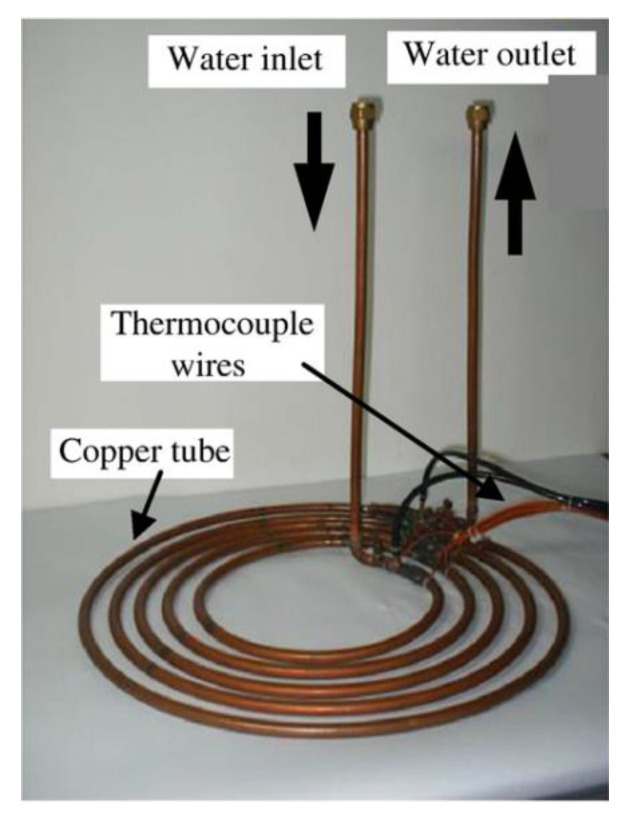
The physical model of spiral-coil tube.

**Figure 5 micromachines-13-01206-f005:**
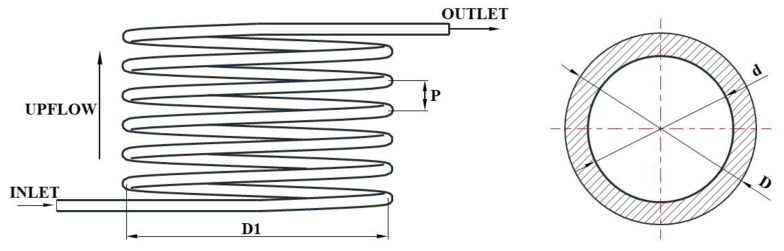
Schematic of spiral tube geometry model.

**Figure 6 micromachines-13-01206-f006:**
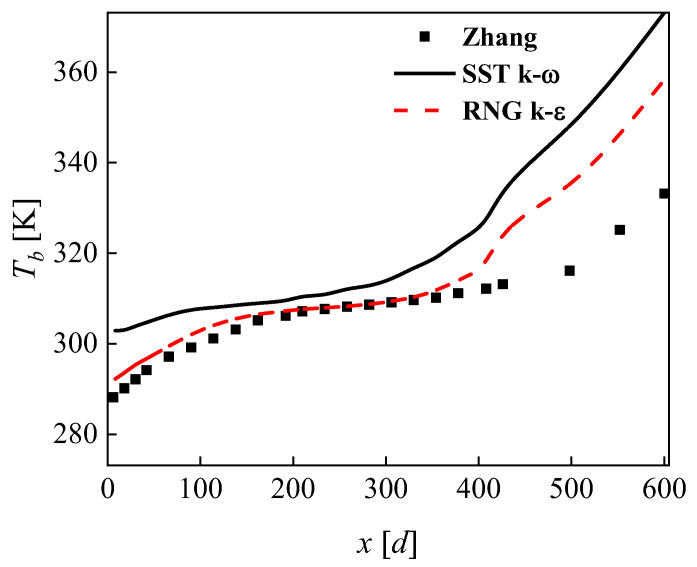
Comparison of simulation and experimental values of the average temperature of the inner cross-section of the spiral tube.

**Figure 7 micromachines-13-01206-f007:**
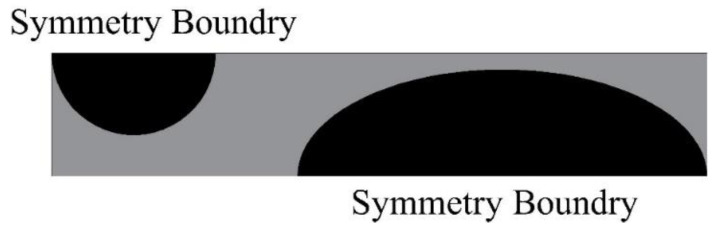
Heat exchanger unit boundary conditions.

**Figure 8 micromachines-13-01206-f008:**
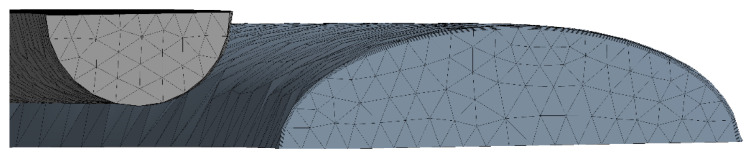
Grid partition.

**Figure 9 micromachines-13-01206-f009:**
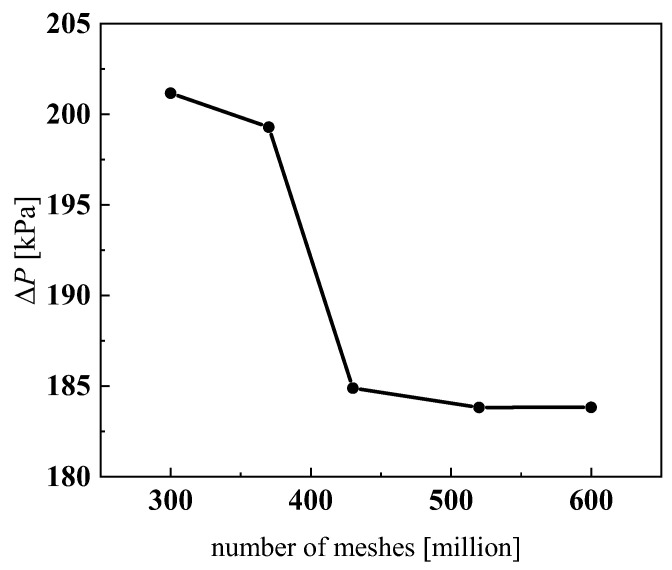
Grid independence.

**Figure 10 micromachines-13-01206-f010:**
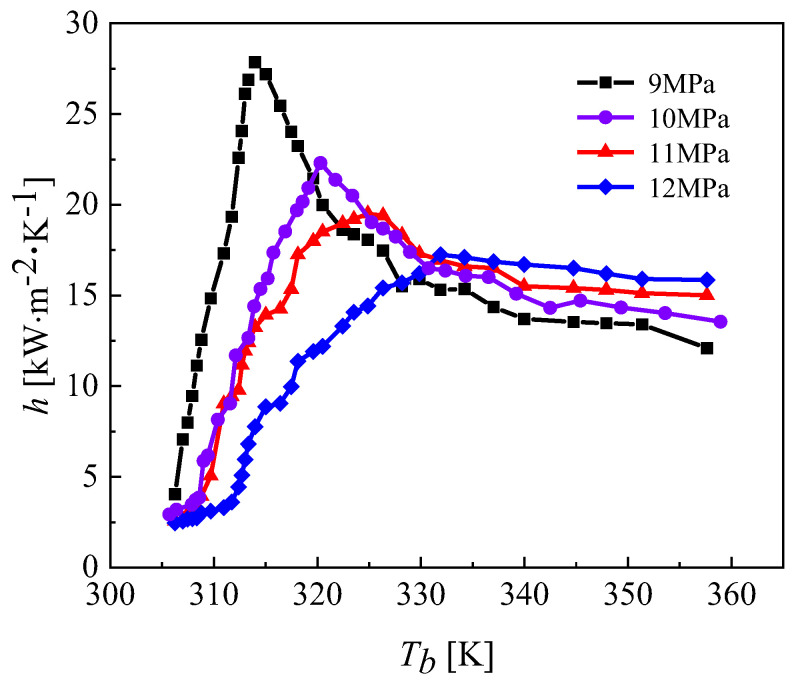
Effect on the heat transfer coefficient of CO_2_ at different cooling pressures.

**Figure 11 micromachines-13-01206-f011:**
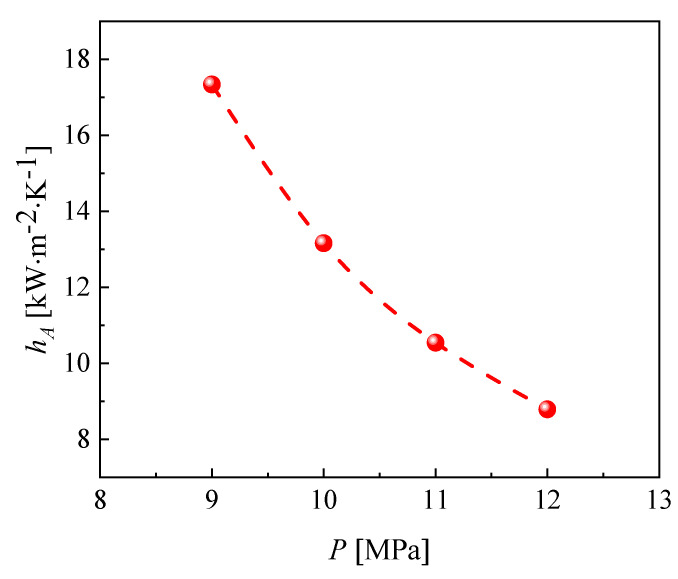
Variation in the average heat transfer coefficient of CO_2_ at different cooling pressures.

**Figure 12 micromachines-13-01206-f012:**
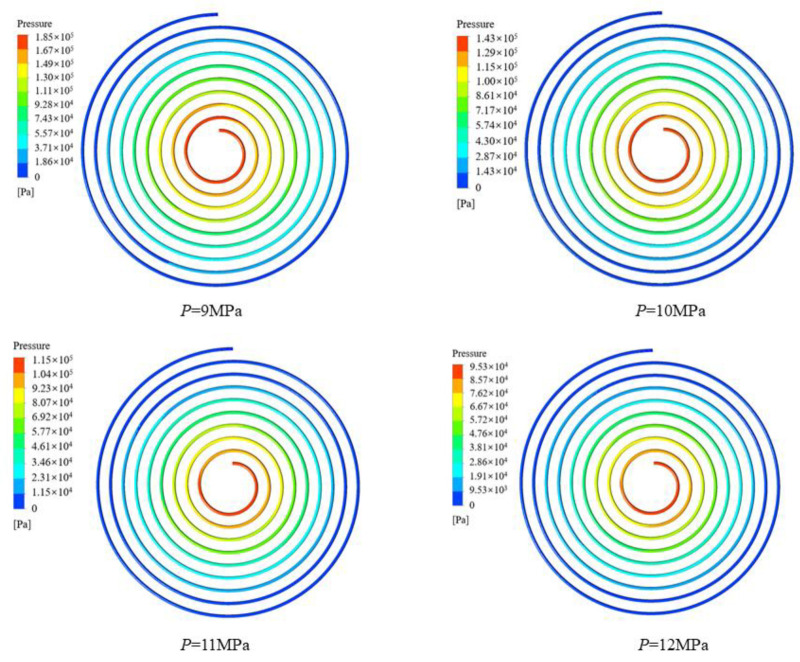
Variation in the pressure contour of CO_2_ at different cooling pressures.

**Figure 13 micromachines-13-01206-f013:**
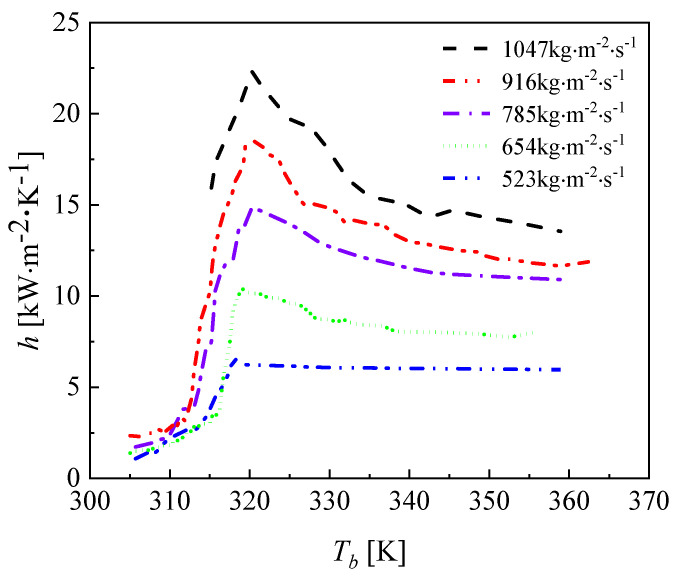
Variation in the heat transfer coefficient of CO_2_ at different mass fluxes.

**Figure 14 micromachines-13-01206-f014:**
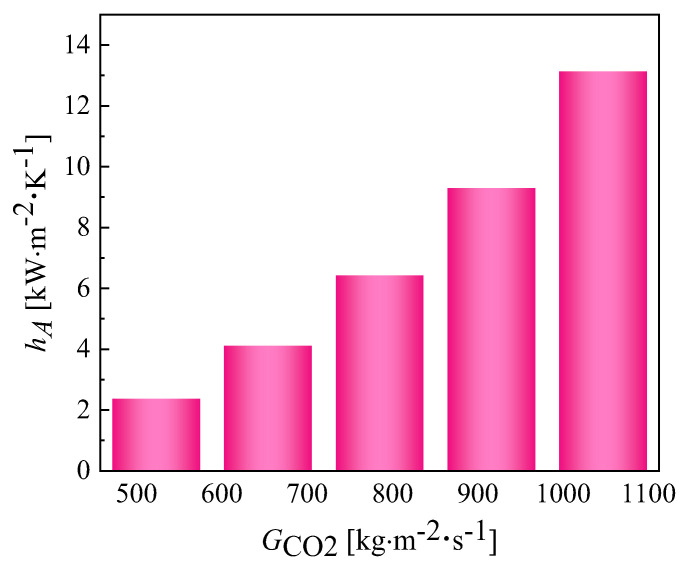
Variation in the average heat transfer coefficient of CO_2_ at different mass fluxes.

**Figure 15 micromachines-13-01206-f015:**
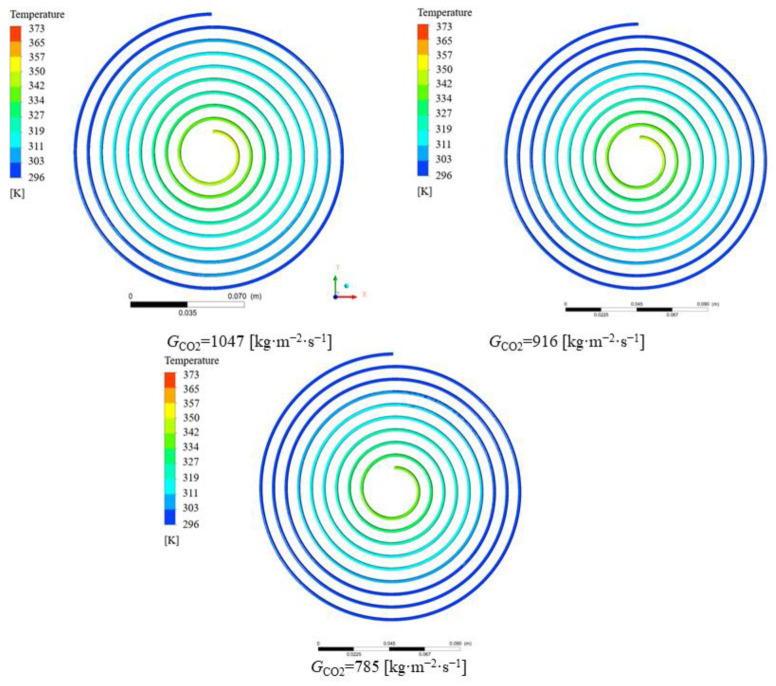
Variation in the temperature of CO_2_ at different mass fluxes.

**Figure 16 micromachines-13-01206-f016:**
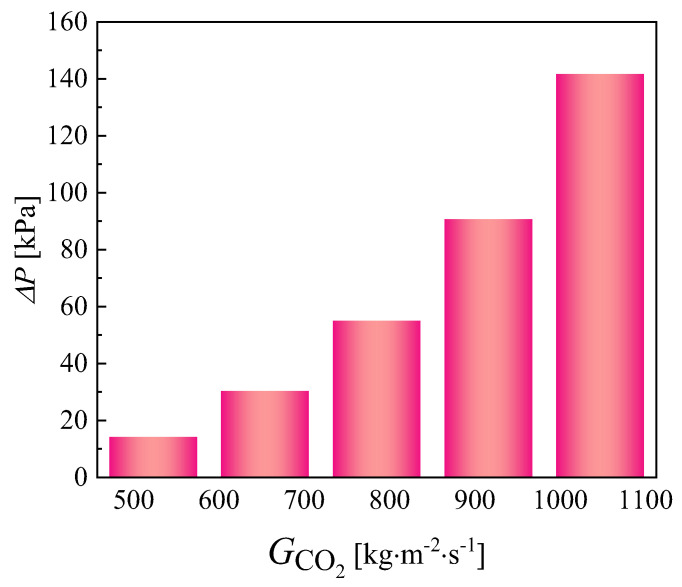
Variation in the pressure drop of CO_2_ at different mass fluxes.

**Figure 17 micromachines-13-01206-f017:**
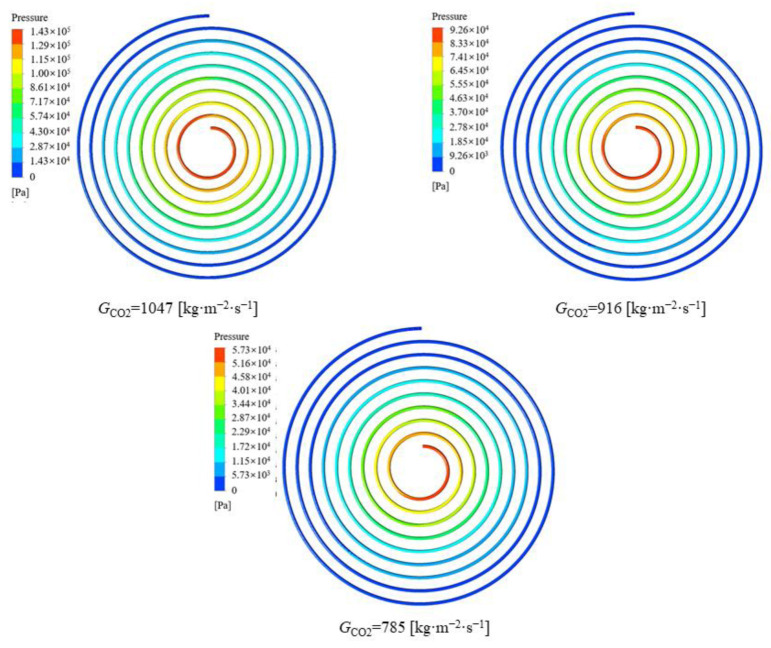
Variation in the pressure contour of CO_2_ at different mass fluxes.

**Figure 18 micromachines-13-01206-f018:**
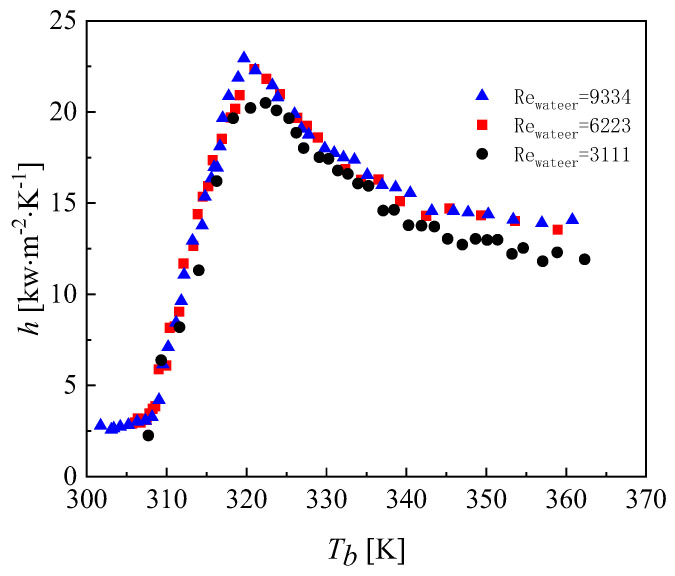
Variation in the heat transfer coefficient of CO_2_ at different water mass fluxes.

**Figure 19 micromachines-13-01206-f019:**
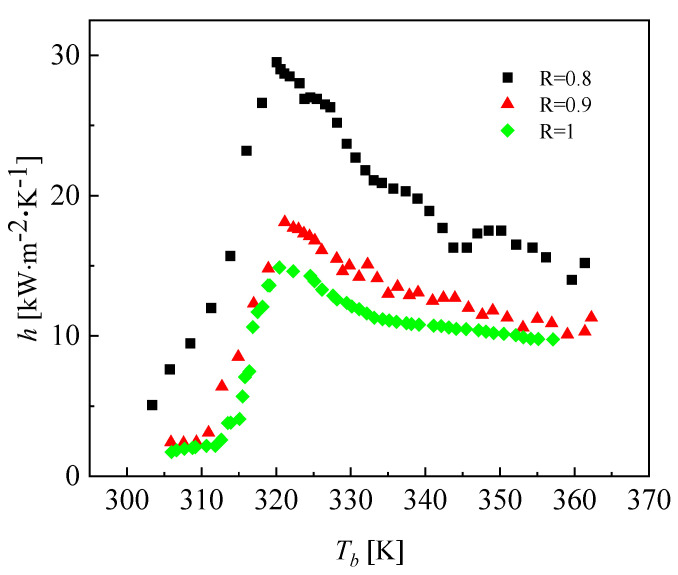
Effect of channel radius on the heat transfer coefficient of CO_2_.

**Figure 20 micromachines-13-01206-f020:**
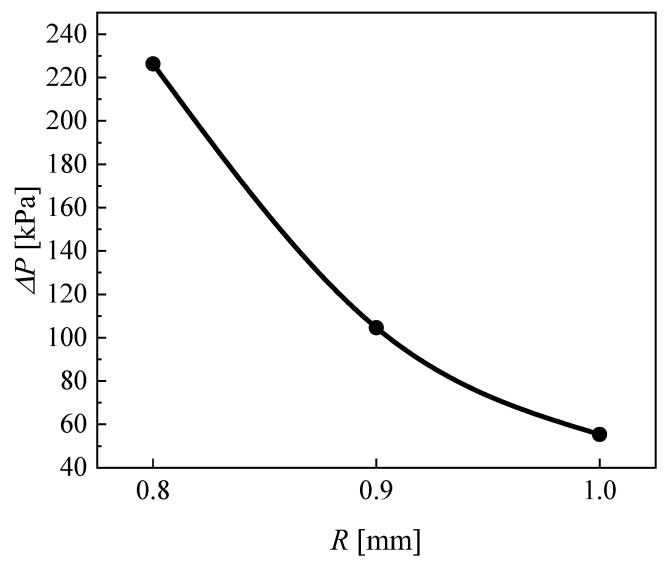
Effect of channel radius on the pressure drop of CO_2_.

**Figure 21 micromachines-13-01206-f021:**
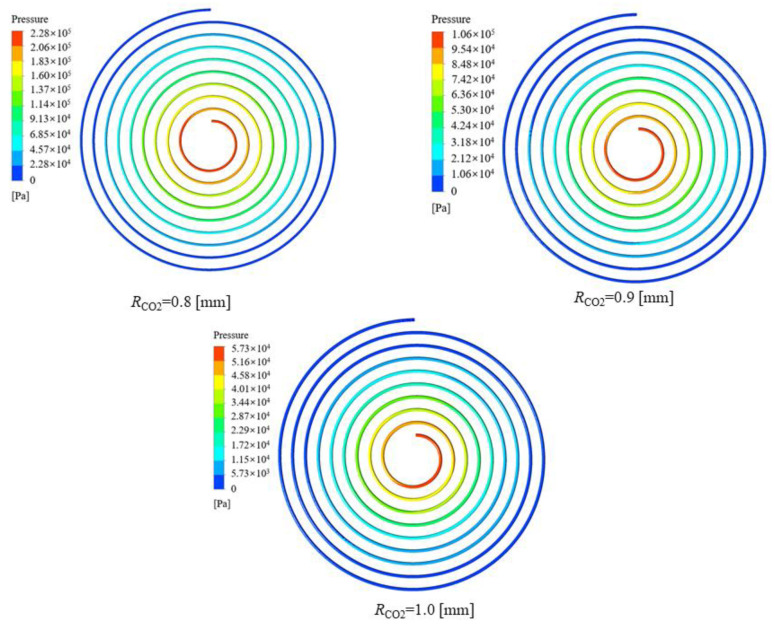
The pressure contour of CO_2_ at different channel radiuses.

**Figure 22 micromachines-13-01206-f022:**
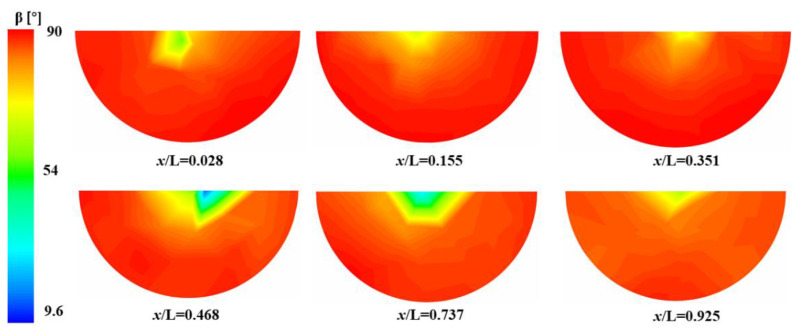
The variation in the synergy angle in different axial positions.

**Table 1 micromachines-13-01206-t001:** Important dimensions of the gas cooler.

Overall Dimensions	
Gas cooler length	200 mm
Gas cooler width	200 mm
Gas cooler height	270 mm
Gas cooler single layer plate thickness	1.5 mm
**Carbon Dioxide Side**	
Carbon dioxide channel length	3.2 m
Carbon dioxide channel radius	1 mm
**Water Side**	
Elliptical channel length	2.8 m
Elliptical channel short semi-axis	1.3 mm
Elliptical channel long semi-axis	2.5 mm

**Table 2 micromachines-13-01206-t002:** Calculation conditions.

Case	*G*_CO2_ (kg·m^−2^·s^−1^)	*P*_water_ (MPa)	*T*_water_ (K)	*P*_CO2_ (MPa)	*T*_CO2_ (K)
1	1047	0.5	293.15	9	373.15
2	1047	0.5	293.15	10	373.15
3	1047	0.5	293.15	11	373.15
4	1047	0.5	293.15	12	373.15

**Table 3 micromachines-13-01206-t003:** Calculation conditions.

Case	*G*_CO2_ (kg·m^−2^·s^−1^)	*P*_water_ (MPa)	*T*_water_ (K)	*P*_CO2_ (MPa)	*T*_CO2_ (K)
1	523	0.5	293.15	10	373.15
2	654	0.5	293.15	10	373.15
3	785	0.5	293.15	10	373.15
4	916	0.5	293.15	10	373.15
5	1047	0.5	293.15	10	373.15

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
