# Peer review of "Study of New Mini-Channel Trans-Critical CO2 Heat Pump Gas Cooler"

_micromachines, 2022, doi:10.3390/mi13081206_

Round 1

Reviewer 1 Report

To reach the standards of the journal, some changes have definitely to be made. Have a look at the following hints. 

- In the abstract,
a) line 15  - a capital letter is missing after a full stop:
b) line 17 - a spiral channel HE is "designed"? Please change. That is not clear:
c) The sentence "The results show that the cooling pressure of CO2, mass flux of CO2 and channel 20 radius of CO2 have a large influence on the local heat transfer coefficient" is obvious. Please tell briefly the main results and the main hypotheses to interpret them;

- Literature review:
Either new or different models and ways to simulate and inspect heat exchangers local behavior should be part of the references. The following could be hints to be included in the references and briefly commented in the paper:
a) A hybrid method for the cross flow compact heat exchangers design Applied Thermal Engineering, 2017, 111, pp. 1129–1142
b) Sensitivity Analysis of Evaporative Condensers Performance Using an Experimental Approach Energy Procedia, 2017, 126, pp. 345–352
c) Numerical and Experimental Performance Analysis of Evaporative Condensers  Energy Procedia, 2016, 101, pp. 26–33
d) A comprehensive design and rating study of evaporative coolers and condensers. Part I. Performance evaluation - (2006) International Journal of Refrigeration, 29 (4), pp. 645-658.
e) A comprehensive design and rating study of evaporative coolers and condensers. Part II. Sensitivity analysis (2006) International Journal of Refrigeration, 29 (4), pp. 659-668.
f) Multiscale Computational Fluid Dynamics Methodology for Predicting Thermal Performance of Compact Heat Exchangers  (2016) Journal of Heat Transfer, 138 (7), art. no. 071801.

 - nothing is said about the sealing. How the design can avoid the mix of the fluids (water and CO2) given the highly different pressures;

- Governing equation can be moved to an Appendix;

- nothing is said about the heat transfer through the metal part separating the fluxes, its temperature distribution and its thermal inertia (if considered in non-stationary phemomena;

- both in x- and y-axis of figures, the unit is reported after a "/". The right notation is using square brackets for units. Please change everywhere;   

- a local "h" investigation with the spiral channel length reached could be interesting to the reader (if I understood that local h was calculated ) as the average h is an information similar to the transferred heat.

Sincerely 
the reviewer

Author Response

  1. line 15 - a capital letter is missing after a full stop:

Response: Thank you very much for your recommendation. We have modified to capital letters.

  1. line 17 - a spiral channel HE is "designed"? Please change. That is not clear:

Response: Thanks for your kind suggestions. We have changed it to “A heat exchanger with spiral channel is studied”.

  1. The sentence "The results show that the cooling pressure of CO2, mass flux of CO2 and channel 20 radius of CO2 have a large influence on the local heat transfer coefficient" is obvious. Please tell briefly the main results and the main hypotheses to interpret them;

Response: We are grateful for the suggestion. We have added brief conclusions and the scope of the study.

  1. Literature review

Response: Thanks for the references, which are now included in the revised manuscript. Either new or different models and ways to simulate and inspect heat exchangers local behavior have been added in the references, literature review has been briefly commented in the paper.

  1. nothing is said about the sealing. How the design can avoid the mix of the fluids (water and CO2) given the highly different pressures;

Response: Thanks for your suggestions. We have inserted the relevant narrative in section 2.

  1. Governing equation can be moved to an Appendix

Response: Thanks for your comments. Governing equation has been moved to Appendix.

  1. nothing is said about the heat transfer through the metal part separating the fluxes, its temperature distribution and its thermal inertia (if considered in non-stationary phemomena.

Response: We are extremely grateful to reviewer for pointing out this problem. The paper is mainly to study the stationary condition, to lay the foundation for engineering applications, the main concern is the heat transfer between the fluids. There is a heat exchanger is being processed and manufactured, the subsequent experimental research can be carried out to study the non-stationary process and the research is more accurate.

  1. both in x- and y-axis of figures, the unit is reported after a "/". The right notation is using square brackets for units. Please change everywhere

Response: Thanks for your comments. The x- and y-axis of all figures have been modified.

  1. a local "h" investigation with the spiral channel length reached could be interesting to the reader (if I understood that local h was calculated) as the average h is an information similar to the transferred heat.

Response: Thanks for your comments. Figure 10, figure 12 and figure 15 display the magnitude and distribution of the local heat transfer coefficients, figure 18 shows the distribution of the along-range synergy angle, which can also indicate the local heat transfer.

Reviewer 2 Report

In order to estimate the CO2 heat transfer behaviour in a spiral plate heat exchanger, CFD analyses are carried out in this research and corresponding results are discussed. However, the innovation of the paper in present version is still lacking. Some suggestion as follow can be referred.

1. To show respect for the pioneer of the transcritical CO2 technology, Prof. Lorentzen rather than other researchers should be mentioned, especially in citation [1].

2. The innovation of this research is not very clear. What are the shortcomings of relevant studies by other scholars? And what innovative method does this paper use to solve what problem? 

3. The units are missing in Fig. 3. Besides, how can it be reflected from Fig. 3 that the elliptical channel can withstand pressures above 20MPa?

4. How to verify the accuracy of analysis results from section 3?

5. The citation format of the reference [14] is incorrect.

6. In section 5.1~5.3, as a research based on CFD simulation, whether the change of temperature field and pressure field in the calculation domain with the change of conditions should be given, so as to intuitively reflect the relevant conclusions? 

7. In section 5.1, the author is strongly suggested to provide more deep analysis and discussion rather than only describing the curves in the figures.

8. How to verify the accuracy of analysis results from section 5? Because the differences between CFD and experimental results are relatively high. Besides, how about the verification in CFD results of pressure drop?

9. The innovation of this paper is not sufficient. For example, the conclusions such as "mass flow rate increases, heat transfer coefficient increases and pressure drop increases" can be expected by all the workers in this field. The author only calculated some specific data in a specific structure, which can be said to be only some obvious conclusions.  Therefore, the innovation aspect of this paper needs to be strengthened. 

Reviewer 3 Report

My review is attached in pdf file.

Round 2

Reviewer 1 Report

Authors made the requested changes.

Sincerely

the reviewer

Author Response

Thanks to the reviewer.

Reviewer 2 Report

1. In question 4 of the response to reviewer 2, the reviewer did not understand what the authors are trying to say. The reviewer's qurstion was, whether the theoretical CFD results in Section 3 were validated?

2. The author states that this paper adopted a new analysis method, which was not well explained. The reviewer did not find any novel research or analysis methods, and believed that the traditional CFD analysis method was adopted in this paper.

3. The reviewer agree with the author's statement that this paper carried out correlation analysis for a new structure and obtained certain theoretical results. However, the experimental comparison shown in Figure 6 alone was far from illustrating the accuracy of all simulation results.

4. Reviewers agreed with the authors that studies that focused on engineering applications needed to sort out regularity from CFD results.  However, the reviewer suggested that some CFD results should be added, for example, if pressure loss was studied in this paper, then at least some pressure contours should be provided. 
